A new species of Brachycephalus (Anura: Brachycephalidae) from southern Brazil

http://orcid.org/0000-0002-2949-4871 Pie Marcio R. 1 2
Ribeiro Luiz F. 2 3
Confetti André E. 4
Nadaline Mário J. 1
Bornschein Marcos R. 2 5 bornschein.marcao@gmail.com
1 Departamento de Zoologia, Universidade Federal do Paraná , Curitiba, Paraná , Brazil
2 Mater Natura—Instituto de Estudos Ambientais , Curitiba, Paraná , Brazil
3 Escola de Ciências da Vida, Pontifícia Universidade Católica do Paraná , Curitiba, Paraná , Brazil
4 Programa de Pós-Graduação em Zoologia, Universidade Federal do Paraná , Curitiba, Paraná , Brazil
5 Instituto de Biociências, Campus do Litoral Paulista, Universidade Estadual Paulista , São Vicente, São Paulo , Brazil
Oehlmann Jörg
Electronic publication date: 2018 Oct 3
Publication date: 2018
Volume: 6
Electronic Location ID: e5683
Received 2018 Apr 18; Accepted 2018 Aug 20
Copyright: © 2018 Pie et al.
Copyright year: 2018
Copyright holder: Pie et al.
License: This is an open access article distributed under the terms of the Creative Commons Attribution License, which permits unrestricted use, distribution, reproduction and adaptation in any medium and for any purpose provided that it is properly attributed. For attribution, the original author(s), title, publication source (PeerJ) and either DOI or URL of the article must be cited.
License URL: https://creativecommons.org/licenses/by/4.0/

Keywords: Atlantic forest, Montane forest, Serra do Mar, Brachycephalus pernix group, Linea masculina, Note-centered approach, Brachycephalus mirissimus, Conservation

Funding: CNPq/MCT 301636/2016-8 André Eduardo Confetti through a grant from CAPES Fundação Grupo Boticário de Proteção à Natureza, through a project A0010_2014 Conducted by Mater Natura—Instituto de Estudos Ambientais Marcio Roberto Pie was supported through a grant from CNPq/MCT (301636/2016-8) and André Eduardo Confetti through a grant from CAPES. Fieldwork was funded by Fundação Grupo Boticário de Proteção à Natureza, through a project (A0010_2014) conducted by Mater Natura—Instituto de Estudos Ambientais. The funders had no role in study design, data collection and analysis, decision to publish, or preparation of the manuscript.

==============================
A new miniaturized frog of the genus Brachycephalus (Anura: Brachycephalidae) is described from Morro Santo Anjo in the municipality of Massaranduba, Santa Catarina, southern Brazil. Specimens were collected from the leaf litter between 470 and 540 above sea level. The new species is distinguished from all its congeners by the combination of the following characters: (1) body robust and bufoniform; (2) size snout-vent length 9.9–11.7 mm for males and 10.0–12.9 mm for females; (3) smooth dorsum; (4) general color (in life) orange with white dots and stripe in the middle of the head and along its vertebral column; (5) iris completely black; (6) advertisement call composed of note groups; (7) isolated notes with 1–3 pulses; and (8) short isolated notes (0.002–0.027 s). An estimate of the male density of the new species is also presented. Phylogenetic information indicates that the new species is part of the southernmost clade of Brachycephalus, which includes Brachycephalus fuscolineatus, B. albolineatus, and B. boticario. The severe anthropogenic impacts in and around the type locality indicate that immediate actions should be taken to ensure the long-term preservation of the new species.

Introduction

Although the first Brachycephalus was discovered in the early 19th century, half of its 35 currently recognized species have been described since 2011 (Frost, 2018). One of the main reasons for this late burst of new species descriptions is probably the relative inaccessibility of many of its populations, which tend to be restricted to high elevation regions of the Brazilian Atlantic Forest (Pie et al., 2013; Bornschein et al., 2016a). Many of these species are microendemic, being found only in one or a few adjacent mountaintops (Bornschein et al., 2016a), a factor that, together with the destruction of their habitat and their relatively low reproductive rate, makes them particularly vulnerable to extinction. Therefore, there is an urgent need to advance our understanding of the taxonomy and distribution of Brachycephalus species to ensure their long-term conservation.

Brachycephalus has been recently divided into three phenetic groups, namely the Brachycephalus ephippium, B. didactylus, and B. pernix species groups (Ribeiro et al., 2015). There are intriguing differences in how species of each group respond to altitude. For instance, species from the B. didactylus group occur from sea level to higher altitudes (0–1,110 m a.s.l.) and are more environmentally tolerant (Pie et al., 2013; Bornschein et al., 2016a). On the other hand, species from the remaining groups tend to be found at higher elevations, but can also occur locally at lower altitudes if particular microclimatic conditions are met (B. ephippium group occurs between 200 and 1,900 m a.s.l. and B. pernix group between 455 and 1,640 m a.s.l.; Bornschein et al., 2016a). Therefore, it is not surprising that extensive field work in the highlands of the southern Brazilian Atlantic Forest led to the discovery of several new species of Brachycephalus of the B. pernix group (Pie & Ribeiro, 2015; Ribeiro et al., 2015; Bornschein et al., 2016b; Ribeiro et al., 2017). As part of this continued effort, in the present study, we describe a new species of Brachycephalus from the state of Santa Catarina, southern Brazil.

Methods

Procedures with specimens and specimens examined

Collection permits for this study were issued by ICMBIO (10.500, 55918–1). Collected specimens were anaesthetized and euthanized using 2% chloridrate lidocaine, fixed in 10% formalin, stored in 70% ethyl alcohol solution, and deposited in the Museu de História Natural Capão da Imbuia (MHNCI), Curitiba, state of Paraná, Brazil. We also examined specimens deposited in the following Brazilian collections: Célio F. B. Haddad collection, Departamento de Zoologia, Universidade Estadual Paulista, Campus de Rio Claro, state of São Paulo (CFBH); MHNCI; Coleção Herpetológica do Departamento de Zoologia, Universidade Federal do Paraná, Curitiba, state of Paraná (DZUP); Museu Nacional, Rio de Janeiro, state of Rio de Janeiro (MNRJ); Museu de Zoologia da Universidade de São Paulo, São Paulo, state of São Paulo (MZUSP); and Museu de História Natural, Universidade Estadual de Campinas, Campinas, state of São Paulo (ZUEC). A list of the examined specimens is provided in the Appendix 1.

Morphometric measurements were obtained with a micrometric eyepiece attached to a Zeiss Stemi 2000 stereomicroscope. Abbreviations for the 15 measurements used according to Watters et al. (2016) were as follows: snout-vent length (SVL), distance from tip of snout to opening of the cloaca; head length (HL), from tip of snout to angle of jaw; head width (HW), greatest width of head located between angles of jaw; eye diameter (ED), from the anterior to posterior corner of the eye; interorbital distance (IOD), between anterior corners of the eyes; internarial distance (IND), shortest distance between inner margins of nostrils; eye–nostril distance (EN), from anterior corner of the eye to posterior margin of nostril; snout length (SL), distance from the tip of the snout to the anterior corner of the eye; upper eyelid width (UEW), greatest width of the upper eyelid margins, measured perpendicular to the anterior–posterior axis; forearm length (FLL), from the flexed elbow to the base of the outer palmar tubercle; hand length (HAL), from the base of the outer palmar tubercle to the tip of finger IV; thigh length (THL), distance from the opening line of the cloaca to the knee; tibia length (TL), distance from the outer surface of the flexed knee to the heel; tarsus length (TSL), from the tibiotarsal articulation to the base of the inner metatarsal tubercle; foot length (FL), from the base of inner metatarsal tubercle to the tip of toe IV. All measurements are indicated in millimeter and were obtained by a single researcher (Luiz Fernando Ribeiro (LFR)).

The sex of the specimens was determined by the presence of the linea masculina, which is only found in males of some anurans species, and consists of bands of fibrous connective tissue located over the entire extension of the oblique muscles (Duellman & Trueb, 1986). The linea masculina can be easily seen when the skin of the ventral region is cut (Ribeiro et al., 2017), or, also, occasionally by transparency in specimens with light ventral color (see below). This character is present in all species of the B. pernix group (Ribeiro et al., 2017), in species of the B. didactylus group (at least in B. hermogenesi) and is absent in species of the B. ephippium group (at least in B. vertebralis, B. pitanga, and B. toby; LFR & Marcos Ricardo Bornschein (MRB), 2017, personal observation). We assigned the new species into one of the three phenetic species groups, considering species with bufoniform body shape and with no linea masculina as belonging to the B. ephippium species group, species with bufoniform body shape and linea masculina as belonging to the B. pernix species group, and species with leptodactyliform body shape and linea masculina as belonging to the B. didactylus species group (as above; modified from Ribeiro et al. (2017)).

Molecular phylogeny

To determine the phylogenetic position of the new species within the species group, we sequenced one of the paratypes (MHNCI 10798). Whole genomic DNA was extracted using PureLink™ Genomic DNA kit (Invitrogen™, Carlsbad, CA, USA), according to the manufacturer’s instructions. One mitochondrial locus (16S rRNA) was amplified by polymerase chain reaction (PCR). PCR was performed in a final volume of 25 μL and consisted of 2 U AmpliTaq DNA polymerase, 1× PCR buffer, 1.5 mM MgCl2, 0.5 mM dNTPs, 1.0 μM each primer (16SA-L CGCCTGTTTATCAAAAACAT and 16SB-H CCCGTCTGAACTCAGATCACGT; Vences et al., 2000) and approximately 30 ng of template DNA. Thermocycling conditions involved an initial denaturation at 94 °C for 5 min, followed by 35 cycles at 94 °C for 1 min, 56 °C for 50 s and 72 °C for 1 min; and a final extension at 72 °C for 5 min. PCR products were electrophoresed on 1.5% agarose gels, and positive PCR products were purified using PEG 8000. Sequencing reactions were performed in a final volume of 10 μL, consisting of 0.7 μL ABI Prism® BigDye™ v3.1 (Applied Biosystems Inc., Foster City, CA, USA), 1.0 μL 5× buffer and one μL each (3.2 pmol) primer and approximately 30 ng of template DNA. Cycle sequencing conditions included an initial denaturation step of 96 °C for 1 min, followed by 35 cycles of 15 s at 96 °C for denaturation, 15 s of annealing at 50 °C and extension of 4 min at 60 °C. Each locus was sequenced in both directions, and sequencing was performed in an ABI 3500 sequencer. The obtained sequences were aligned with all available 16S sequences of species of the B. pernix group on GenBank, as well as one sequence of B. didactylus as the outgroup (Table S1) using MUSCLE v3.8.31 (Edgar, 2004) under default settings. Regions with ambiguous alignments were omitted from the final analyses. A phylogeny was obtained using a Bayesian approach with MrBayes 3.2 (Ronquist et al., 2012) after the best model of evolution was determined using jModelTest 2.1.7 (Darriba et al., 2012) to be HKY+ Γ. Each analysis consisted of two independent runs, each with four chains, run for 5 × 107 generations with sampling every 1,000th generation. After ensuring convergence of separate chains, data sets were combined. Stationary distribution and effective sample sizes (ESS) for all parameters were checked using Tracer v1.5 (Rambaut & Drummond, 2009). We disregarded the initial 20% of the trees as burn-in, and using the remaining trees we estimated the maximum clade credibility consensus topology in TreeAnnotator v1.7.5 (Drummond & Rambaut, 2007; Drummond et al., 2012). The used alignment and MrBayes command block are available in File S1.

Advertisement call description

We recorded specimens of the new species on January 15 and 18, 2018 at the type locality of the species (see the Results). Recordings were carried out from 9.00 to 12.00 a.m. Climatic conditions during recordings were characterized by air temperature = 20.4–25.8 °C, soil temperature = 19.1–23.2 °C, and relative air humidity = 100%. Calls were recorded using the digital recorders Sony PCM-D50 with a Sennheiser ME 66/K6 microphone, Marantz PMD660 with a Sennheiser ME 66/K6 microphone, and Tascam DR44-WL with a Sennheiser ME 67/K6 microphone, all with sampling frequency rate of 44.1 kHz and 16-bit resolution. Recordings were deposited at MHNCI. Sound samples were analyzed with Raven Pro 1.5 (Bioacoustics Research Program, 2012). Time domain variables were measured from oscillograms and frequency domain variables were measured from spectrograms. Spectrogram features were defined with a 256-point Fast Fourier Transform and a 3-dB Filter bandwidth of 492 Hz, Hann window, 50% overlap. Spectrograms for figures, as well as diagnostic plots, were generated using the Seewave package, v. 2.0.5 (Sueur, Aubin & Simonis, 2008) of the R environment, v. 3.2.2 (R Core Team, 2016) and overlap settings as in Raven Pro.

We described the advertisement calls following features, criteria, and note-centered approach of Köhler et al. (2017) and particularities as in Bornschein et al. (2018). We considered the end of a given call and the beginning of the next one by the long period of silence between them (Köhler et al., 2017), which might last for several minutes and thus is considerably longer than the call itself. We used the following features (see Fig. 1 of Bornschein et al. (2018)): (1) call duration (s); (2) duration of the call including only isolated notes (s); (3) duration of the call including only note groups (s); (4) note rate (notes per minute); (5) note rate of the call including only isolated notes (notes per minute); (6) note rate of the call including only note groups (notes per minute); (7) number of notes per call; (8) number of isolated notes per call; (9) number of note groups per call; (10) number of pulses per isolated notes; (11) number of pulses per note in note groups; (12) note duration of isolated notes (s); (13) duration of note group (s); (14) inter-note interval in isolated notes (s), defined as the time from the end of one isolated note to the beginning of the next note isolated note; (15) inter-note group interval (s), defined as the time from the end of one note group to the beginning of the next note group; (16) inter-note interval within note groups (s), defined as the time from the end of the first note to the beginning of the next note of the same note group; (17) note dominant frequency (kHz); (18) highest frequency (kHz); and (19) lowest frequency (kHz). The note rate was calculated taking into account the time from the beginning of the first note to the beginning of the last note of the calls (or call intervals) and the number of notes included in this counted time (Bornschein et al., 2018). The highest and lowest frequencies we measured from notes. Some calls began with notes with very reduced range of frequency in relation to subsequent notes. These “warming” notes (sensu Bornschein et al., 2007) were not included in the calculation of the parameters 12 and 17–19, above. We compared the advertisement calls of the new species with calls descriptions in literature, as well with calls deposited in public institutions, namely MHNCI and Fonoteca Neotropical Jacques Vielliard (FNJV) (Appendix 2).

Counting calling males

We obtained an approximate estimate of male density following the methods indicated in Bornschein et al. (2016a) and Ribeiro et al. (2017). We slowly walked along a small trail that crossed the study area as a transect in 3 days. We than select, in the third day, the point where the species appeared to be abundant and spent approximately 5 h placing markings on the vegetation at the positions where we recorded a calling male. We then measured the extent of the sampling area (= the extent of the auditory sampling) and counted the number of markings, resulting in an estimative of calling males per area. We used the classification of Brazilian vegetation proposed by the RADAMBRASIL project (in Veloso, Rangel-Filho & Lima, 1991) to characterize the habitats of the species. Altitudinal records were obtained after plotting the geographical coordinates of the lowest and highest altitudinal records in the field using Google Earth.

Estimation of area of occurrence

We estimate the “actual” area or occurrence of the species by considering all areas within the lowest altitudinal record of the species up to the highest altitude in the place where it was recorded, excluding deforested areas and other areas eventually without records (Bornschein et al., 2016a). If the species shows a potentially uniform distribution area of occurrence, we will consider it as “area of occupancy” (sensu IUCN, 2012); if the species showed a patchy occurrence pattern, we considered it area of occurrence as “extent of occurrence” (sensu IUCN, 2012; Bornschein et al., 2016a).

Registration of nomenclatural act

The electronic version of this article in portable document format will represent a published work according to the International Commission on Zoological Nomenclature (ICZN), and hence the new name contained in the electronic version are effectively published under that Code from the electronic edition alone. This published work and the nomenclatural acts it contains have been registered in ZooBank, the online registration system for the ICZN. The ZooBank LSIDs (Life Science Identifiers) can be resolved and the associated information viewed through any standard web browser by appending the LSID to the prefix http://zoobank.org/. The LSID for this publication is: urn:lsid:zoobank.org:pub:AAB0DCA9-4587-42B3-812E-7752EB58F726. The online version of this work is archived and available from the following digital repositories: PeerJ, PubMed Central and CLOCKSS.

Results

Brachycephalus mirissimus sp. nov.

Urn:lsid:zoobank.org:act:urn:lsid:zoobank.org:act: urn:lsid:zoobank.org:act:B1C3F142-E68C-4B4F-9647-F69CA6C25EB9

Figures 1–4

Figure 1 Holotype of Brachycephalus mirissimus (MHNCI 10793), adult male, in preservative in dorsal (A) and (B) ventral view of the body.

White bar = 5 mm. Photographs by Luiz F. Ribeiro.

Figure 2 Drawings of the holotype of Brachycephalus mirissimus.

Holotype of Brachycephalus mirissimus (MHNCI 10793), adult male: (A) dorsal view of the body; (B) lateral view of the head; (C) ventral view of right hand; (D) ventral view of right foot. Drawing by Verônica R. Apolônio.

Figure 3 Holotype of Brachycephalus mirissimus (MHNCI 10793), adult male, in life in lateral (A), dorsal (B) and ventral view (C).

Notice in C, from the left to right, two white arrows indicated the presence of the linea masculinea, a pair of bands of fibrous connective tissue present only in males of the species and the vocal sac. Black bar in C = 5 mm. Photographs by Luiz F. Ribeiro.

Figure 4 Variation in coloration of paratypes of Brachycephalus mirissimus.

(A–F) dorsal view; (G and H) ventral view. (A) MHNCI 10802; (B) MHNCI 10799; (C) MHNCI 10803; (D) MHNCI 10794; (E) MHNCI 10801; (F) MHNCI 10800; (G) MHNCI 10796; (H) MHNCI 10803. Notice in H the white arrow indicated the linea masculinea. Black bar in H = 5 mm. Photographs by Luiz F. Ribeiro.

Holotype. MHNCI 10793 (Figs. 1–3) male, collected at Morro Santo Anjo (26°37′41″S, 48°55′50″W; DATUM WGS84; 535 m a.s.l.), municipality of Massaranduba, state of Santa Catarina, southern Brazil, on January 15, 2018 by Marcos R. Bornschein and Larissa Teixeira.

Paratopotypes. MHNCI 10794 male, collected on January 15, 2018 by Marcos R. Bornschein and Larissa Teixeira; MHNCI 10795–9 and MHNCI 10802–3 males, collected on January 18, 2018 by Marcos R. Bornschein, Marcio R. Pie, Luiz F. Ribeiro, André Confetti, and Mário J. Nadaline; MHNCI 10800–1 females, collected on January 18, 2018 by Marcos R. Bornschein, Marcio R. Pie, Luiz F. Ribeiro, André Confetti, and Mário J. Nadaline.

Diagnosis. B. mirissimus is a member of the genus Brachycephalus based on its position in a phylogenetic tree (Fig. 5). B. mirissimus is a member of the B. pernix group, as defined by Ribeiro et al. (2015) and modified above, by having a bufoniform body shape and linea masculina (Figs. 3C and 4H). B. mirissimus is distinguished from all of the species in the genus by the following combination of characters: (1) body robust and bufoniform; (2) size SVL 9.9–11.7 mm for males and 10.0–12.9 mm for females (Table 1); (3) smooth dorsum (Figs. 3 and 4); (4) general color (in life) orange with white dots and stripe in the middle of the head and along its vertebral column (Figs. 3 and 4); (5) iris completely black (Fig. 3A); (6) advertisement call composed of note groups; (7) isolated notes with 1–3 pulses; and (8) short isolated notes (0.002–0.027 s).

Figure 5 Relationships between species of the Brachycephalus pernix species group based on a partial sequence of the 16S mitochondrial gene.

Phylogenetic analysis was carried out using Bayesian inference and values above branches correspond to node posterior probabilities. Nodes with posterior probabilities lower than 50% were collapsed.

Table 1 Measurements in millimeter of the 15 variables of the type series of Brachycephalus mirissimus.

Variation	Males (n = 9)	Females (n = 2)	
Mean	SD	Range	Mean	SD	Range	
SVL	10.9	0.6	9.9−11.7	11.4	2.0	10.0−12.9	
HL	3.6	0.2	3.3−3.9	3.9	0.5	3.6−4.3	
HW	4.4	0.2	4.0−4.7	4.6	0.6	4.1−5.0	
ED	1.2	0.1	1.1−1.3	1.3	0.1	1.2−1.4	
IOD	2.3	0.1	2.0−2.4	2.3	0.2	2.2−2.5	
IND	1.3	0.1	1.1−1.4	1.3	0.1	1.3−1.4	
EN	0.6	0.0	0.6−0.7	0.7	0.1	0.7−0.8	
SL	0.8	0.1	0.8−1.0	0.9	0.1	0.8−1.0	
UEW	0.8	0.1	0.7−1.0	0.9	0.0	0.9−0.9	
FLL	2.6	0.2	2.3−2.8	2.6	0.4	2.4−2.9	
HAL	1.8	0.1	1.7−2.0	1.9	0.3	1.7−2.1	
THL	4.3	0.2	3.9−4.7	4.5	0.7	4.0−5.0	
TL	3.7	0.2	3.4−4.1	3.9	0.4	3.6−4.1	
TSL	2.8	0.1	2.6−3.0	2.8	0.4	2.5−3.0	
FL	2.9	0.2	2.7−3.2	3.1	0.4	2.8−3.4	
Notes:

See Methods for the corresponding the variable name and abbreviation.

SD, standard deviation.

Comparison between Brachycephalus. B. mirissimus is unique among the species of its genus by its distinctive white dorsal pattern in contrast with an orange body. However, other species present a dorsal stripe, as follows. B. albolineatus have a very similar white dorsal pattern, but not in all specimens and, when present, they also show a green to dark green dorsum. B. boticario, B. auroguttatus, B. mariaeterezae, B. quiririensis, B. guarani (Clemente-Carvalho et al., 2012), some B. ferruginus, and some B. verrucosus also have mid-dorsal stripes but with different coloration: yellow in B. boticario, B. quiririensis, B. auroguttatus, and B. verrucosus, light blue in B. mariaeterezae, brown in B. guarani (Clemente-Carvalho et al., 2012), and reddish-brown in B. ferruginus. The dorsum of some of these species is also different: orange in the new species, as opposed to yellow in B. mariaeterezae, brown in B. quiririensis, light brown in B. boticario, light green in B. verrucosus, brown in B. quiririensis, and pure yellow anteriorly and increasingly mixed with brown instead of yellow toward the posterior region in B. auroguttatus. Two of those species with relatively similar pattern with a contrastingly colored stripe on dorsum have a similar orange dorsum as found in B. mirissimus, that is, B. ferruginus and B. guarani (Clemente-Carvalho et al., 2012). Specimens of B. ferruginus without a reddish-brown stripe on dorsum have dispersed reddish-brown patches on middle dorsum or an entirely orange dorsum. B. mirissimus is also distinguished from some of those species by its coloration on ventral surface, being orange instead of orange with brownish-green regions in B. albolineatus, yellow in B. verrucosus, yellow with small brown spots in B. mariaeterezae, orange with small green spots in B. ferruginus, orange anteriorly and brown with orange spots posteriorly in B. quiririensis, and orange mixed with brown in B. auroguttatus. B. mirissimus is also reminiscent of some specimens of B. fuscolineatus, but nevertheless can be safely distinguished from that species by its orange body and white stripe in contrast with the yellow flanks and dark brown to black middle of the dorsum of B. fuscolineatus. The remaining species of the genus present only one uniform dorsal coloration or two dorsal colorations, but in this case without any of these colors restricted to a narrow stripe, being representatively distributed along dorsum. Beside this, other features can also be used to diagnose the species. From the remaining species of the B. pernix group, B. mirissimus is easily distinguished by the orange body with white dorsal stripe coloration instead of (1) general dorsal body color dark green with dark brown regions in B. actaeus, (2) dark brown in B. brunneus and B. curupira, (3) dark green in B. olivaceus, (4) orange (or orange and yellow), normally with dark spots on the flanks, in B. izecksohni, B. leopardus, B. pombali, and B. tridactylus, (5) orange on head and on central dorsum with black remaining parts in B. pernix, and (6) pale red from the head to the pelvic region and yellowish green on the lateral body and dorsal thighs in B. coloratus. Also, within species of the B. pernix group, the new species share the similar rough dorsum of B. actaeus, B. auroguttatus, B. boticario, B. fuscolineatus, B. leopardus, B. mariaeterezae, B. olivaceus, B. quiririensis, and B. verrucosus, but is distinct in this feature from B. fuscolineatus, that have smooth dorsum. From the remaining species of the B. ephippium group, B. mirissimus is easily distinguished by the orange body with white dorsal stripe coloration instead of (1) general color orange in B. alipioi, B. crispus (Condez et al., 2014), B. darkside (Guimarães et al., 2017), B. ephippium (Pombal, 1999), B. garbeanus (Pombal & Izecksohn, 2011), B. margaritatus (Pombal & Izecksohn, 2011), B. vertebralis, (2) general color orange with dorsal greenish spots in B. toby, (3) general color orange with reddish spots in B. pitanga, and (4) general color greenish yellow to dark gray-green with yellow warts in B. nodoterga (Clemente-Carvalho et al., 2016). B. mirissimus is also distinguishable from species of the B. ephippium group in body size of males, which is larger in some species, such as B. alipioi (SVL = 12.5–16.2 mm; Pombal & Gasparini, 2006), B. darkside (SVL = 14.8–18.5 mm; Guimarães et al., 2017), and B. margaritatus (SVL = 15.0–18.9 mm; Pombal & Izecksohn, 2011). From B. bufonoides, the new species is diagnosable by is body size (SVL 10.0–12.9 mm against 13.5–16.4 in B. bufonoides, considering both sexes; Pombal, 2010). The bufoniform body shape and light color of the body also distinguish the new species from all species of the B. didactylus group, namely B. didactylus (Izecksohn, 1971), B. hermogenesi, B. pulex, and B. sulfuratus, which have leptodactyliform body shapes and homogeneous dorsal coloration, at times with an “X”-shaped darker mark on their dorsum (Izecksohn, 1971; Giaretta & Sawaya, 1998; Napoli et al., 2011; Condez et al., 2016). We were unable to compare the new species with B. atelopoide, from Piquete, São Paulo, because this species is unknown in the wild and its type is apparently missing (Pombal, 2010).

The advertisement calls of B. mirissimus resemble those of species of the B. pernix group, as well as B. hermogenesi, from the B. didactylus group, in that they are composed of relatively short notes, as opposed to the “buzz” structure found in species of the B. ephippium group (see below; see Table S2). With isolated notes having 1–3 pulses, B. mirissimus is easily distinguishable from B. crispus (7–12 pulses per note; Condez et al. (2014)), B. darkside (5–8 pulses per note; Guimarães et al. (2017)), B. ephippium (5–15 pulses per note; Pombal, Sazima & Haddad (1994)), and from B. pitanga (6.9–13.3 pulses per note; Tandel et al. (2014); see also Araújo et al. (2012)), of the B. ephippium group, and is also easily distinguishable from B. sulfuratus (6.90–13.30 pulses per note; Condez et al. (2016)), from the B. didactylus group. The distinctiveness of the new species from those species above can also be expressed by the duration of the isolated notes, short in B. mirissimus (0.002–0.027 s; mean of 0.010 ± 0.007 s) and long in B. crispus (mean of 0.28 ± 0.02 s; Condez et al. (2014)), B. darkside (0.083–0.163 s; Guimarães et al. (2017)), B. ephippium (0.093–0.125 s; Pombal, Sazima & Haddad (1994)), B. pitanga (0.15–0.25 s; Tandel et al. (2014); see also Araújo et al. (2012)), and B. sulfuratus (0.131–0.233 s; Condez et al. (2016)). Within species of the B. pernix group, the advertisement calls of B. mirissimus are easily distinguished from those of B. tridactylus by having note groups, whereas the latter presents only isolated notes. On the other hand, we cannot distinguish the advertisement call of the new species from that of B. albolineatus (Bornschein et al., 2018), due to the general similarity, as least considering the features described to date. Finally, we also cannot distinguish the advertisement call B. mirissimus from that of B. hermogenesi (Verdade et al., 2008) and B. actaeus (Monteiro et al., 2018), but in these cases this is likely due to the simplicity of the description of the call of B. hermogenesi and the use of the call-centered approach in B. actaeus (Table S2) rather than by its supposed similarities.

Description of the holotype. Male with robust bufoniform body; head slightly wider than long; HL 40% of snout-vent length; snout short: its shape semicircular in dorsal view, and rounded in lateral view (Fig. 2); nostrils protuberant, directed anterolaterally; canthus rostralis not distinct; lips nearly sigmoid; loreal region weakly concave; eye slightly protruding in dorsal and lateral views; ED 33% of HL; tympanum indistinct; vocal sac not expanded externally; tongue longer than wide, with posterior half not adherent to floor of mouth; choanae relatively small, rounded; vomerine teeth absent. Upper arm and forearm relatively slender, upper arm approximately as long as forearm; tips of Fingers I, II and II rounded, Finger IV greatly reduced; relative lengths of fingers IV<I<II<III; subarticular tubercles and inner and outer metacarpal tubercles absent; legs short, thigh robust; THL 39% of SVL, crus length 86% of THL; toe II short but distinct, toe III distinct and toe IV long; toes I and V not visible externally; relative length of toes II<III<IV; subarticular tubercles and inner metatarsal tubercles absent; outer metatarsal tubercle distinct, large and ovoid. Skin rough on dorsum of head and central body; skin granular on dorsolateral surfaces of body, flanks, and dorsal surface of thighs, with juxtaposed, large glandular warts; sides of the body granular; large, round juxtaposed glandular warts on the sides of the body, belly and thighs; chin, arms, and legs smooth.

Coloration of the holotype. In life, almost completely orange, except for a patch on the dorsum of the head, a line in the middle of the dorsum and a pair of white patches in the distal dorsum, partly surrounded by a thin yellow margin; dorsal part of leg articulations light orange; iris black (Fig. 3). In preservative, orange regions turned yellowish pale cream and white region (1) remain white or (2) became pale cream, or (3) became pale cream with white edges (Fig. 1).

Measurements of holotype (in mm). SVL = 10.9, HL = 3.7, HW = 4.4, ED = 1.3, IOD = 2.4, IND = 1.3, EN = 0.7, SL = 0.8, UEW = 0.8, FLL = 2.8, HAL = 1.8, THL = 4.3, TL = 4.0, TSL = 2.8, FL = 3.1.

Variation in the type series. Morphometric variation is given in Table 1. There are slight differences in coloration among specimens (Fig. 4). The orange coloration has a yellowish hue in some specimens, both in the dorsal and in ventral regions. In addition, the width of the white stripe and the extent of the white spots on the dorsum of the head and on the distal dorsum also vary (Fig. 1). The latter is light yellow instead of white in some individuals.

Phylogenetic relationships. The phylogenetic analysis of species of the B. pernix species group places B. mirissimus as part a clade which includes B. fuscolineatus and B. boticario (Fig. 5), which are the southernmost species of the genus and are distributed in the region of the new species (Fig. 6).

Figure 6 Map indicating the location of the type locality of Brachycephalus mirissimus (yellow dot), as well as other species of the genus that are found in the region (state of Santa Catarina, southern Brazil) (indicated clockwise): B. mirissimus, B. fuscolineatus, B. boticario and B. albolineatus. Original photographs by Luiz F. Ribeiro. Map data © 2018 Google.

Advertisement call. We analyzed 31 advertisement calls from 12 individuals, six of which were collected (MHNCI 10793–8). We recorded eight individuals two to five times (x¯=3.37 times per individual). The calls we deposited resulted in 31 separate recordings (MHNCI 052–82). All the described features are shown in Table 2 and the distribution of the number of pulses per note are provided in Table 3. Some advertisement calls were not recorded from the beginning, with some notes being heard before the recordings started (Table 3). To measure the duration of the call, we also considered the advertisement calls with up to two initial notes missing in the recordings. Below, we describe the call features reporting the mean ± SD, with the range in parentheses.

Table 2 Measurements of advertisement call features of Brachycephalus mirissimus.

Feature / [analysis]	Mean	SD	Range	n	
Samples	Specimens	
Call duration (s) (1) (entire call)	111.834	46.604	37.700–255.205	25	11	
Duration of the call including only isolated notes (s) (2)	63.055	22.832	16.256–100.304	25	11	
[Duration of the call including only isolated notes when note groups is absent (s)]	67.591	24.175	37.700–96.230	6	4	
[Duration of the call including only isolated notes when note groups occurs (s)]	62.938	22.661	16.256–100.304	19	9	
Duration of the call including only note groups (s) (3)	47.058	39.313	6.706–182.401	22	9	
Note rate (notes per minute) (4) (entire call)	11.694	2.119	7.484–15.935	30	11	
Note rate of the call including only isolated notes (notes per minute) (5)	10.539	1.589	7.484–14.277	30	11	
[Note rate of the call including only isolated notes when note groups is absent (notes per minute)]	9.578	1.978	7.484–12.738	8	5	
[Note rate of the call including only isolated notes when note groups occurs (notes per minute)]	10.809	1.279	8.308–14.227	21	10	
Note rate of the call including only note groups (notes per minute) (6)	17.766	4.658	12.198–27.427	22	9	
Number of notes per call (7) (total notes in entire call)	23.552	10.287	6–52	29	12	
Number of isolated notes per call (8)	13.852	5.340	6–25	27	11	
[Number of isolated notes per call when note groups is absent]	13.286	4.071	7–21	7	4	
[Number of isolated notes per call when note groups occurs]	14.050	5.799	6–25	20	9	
Number of note groups per call (9)	5.133	4.508	0–19	30	12	
[Number of notes in the part of the call including only note groups]	14.000	7.584	6–38	22	10	
Number of pulses per isolated notes (10)	1.691	0.475	1–3	362	12	
Number of pulses per note in note groups (11)	1.958	0.325	1–3	308	10	
[Number of pulses in each note groups]	3.916	0.604	2–6	154	10	
Note duration of isolated notes (s) (12)	0.010	0.007	0.002–0.027	115	10	
Duration of note groups (s) (13)	0.426	0.044	0.361–0.590	56	10	
Inter-note interval in isolated notes (s) (14) (interval between isolated notes)	5.815	1.328	3.919–10.625	84	10	
Inter-note group interval (s) (15) (interval between note groups)	7.022	1.133	5.321–10.930	34	10	
Inter-note interval within note groups (s) (16) (interval between notes in note groups)	0.389	0.030	0.346–0.490	56	10	
Note dominant frequency (kHz) (17)	6.645	0.272	6.000–7.230	227	10	
Highest frequency (kHz) (18)	8.311	0.518	7.143–10.060	227	10	
Lowest frequency (kHz) (19)	4.369	0.767	2.667–5.841	227	10	
Notes:

Numbers in brackets represent the number of the feature in Methods.

SD, standard deviation.

Table 3 Distribution of the number of pulses per note (separated by “,”) along the advertisement calls (AC) of Brachycephalus mirissimus.

n of individuals (voucher number)	Number of pulses per note	Number of isolated notes we hear being emitted before recording the AC	
1 (MHNCI 052)	1, 2, 2, 2, 2, 2, 2	0	
1 (MHNCI 053)	1, 2, 2, 2, 2, 2, 2, 2, 2, 2, 2, 2, 2, (2–2), (2–2), (2–2), (2–2), (2–2), (2–2), (2–2), (2–2), (2–2), (2–2), (2–2), (2–2), (2–2), (2–2), (2–2), (2–2), (2–2), (2–2), (2–2)	0	
2 (MHNCI 054)	2, 2, 2, 2, 2, 2, (2–2), (2–2), (2–2), (2–2), (2–2), (2–2), (2–2)	0	
2 (MHNCI 055)	1, 2, 2, 2, 2, 2, 2, 2, 2, 2, 2, 2, (2–2), (2–2), (2–2), (2–2), (2–2), (2–2), (2–2), (2–2)	0	
3 (MHNCI 056)	2, 2, 2, 2, (2–2), (2–2), (2–2), (2–2), (2–2), (2–2), (2–2), (2–2), (2–2)	2	
4 (MHNCI 057)	1, 1, 2, 2, 2, 2, 2, 2, 2, 2, 2, 2, 2	0	
4 (MHNCI 058)	1, 1, 1, 1, 1, 2, 2, 2, 2, 2, 2, 2, 2	0	
4 (MHNCI 059)	1, 1, 1, 1, 1, 1, 2, 2, 2, 2, 2, 2, 2, 2, 2, 2, 2, 2, 2, (2–2), (2–2), (2–2), (2–2), (2–2), (2–2)	0	
4 (MHNCI 060)	1, 1, 1, 2, 2, 2, 2, 2, 2, 2, 2, 2, 2	0	
5 (MHNCI 061)	1, 1, 1, 1, 1, 1, 2, 2, 2, 2, 2, 2, 2, 2, (2–2), (2–2), (2–2), (2–2), (2–2), (2–2), (2–2), (2–2), (2–2), (2–2)	1	
5 (MHNCI 062)	1, 1, 1, 1, 2, 2, 2, 2, 2, 2, 2, 2, 2, 2, (2–2), (2–2), (2–2), (2–2), (2–2), (2–2), (2–2), (2–2), 1	1	
5 (MHNCI 063)	1, 1, 1, 1, 1, 1, 2, 2, 2, 2, 2, 2, (2–2), (2–2), (2–2), (2–2), (2–2), (2–2), (2–2), (2–2), (2–2), (2–2)	1	
5 (MHNCI 064)	1, 1, 1, 1, 1, 1, 2, 2, 2, 2, 2, (2–2), 2, (2–2), (2–2), (2–2)	1	
5 (MHNCI 065)	1, 1, 1, 1, 2, 2, 2, (2–2), 2, (2–2), (2–2), (2–2), (2–2), (2–2), (2–2), (2–2), (2–2)	2	
6 (MHNCI 066)	1, 1, 1, 1, 2, 2, 2, 2, 2	4	
7 (MHNCI 067)	1, 1, 1, 1, 1, 1, 1, 1, 1, 1, 2, 2, 1, 1, 1, 1, 2, (1–1), (1–1), (1–1), (1–1), (1–1), (1–1), (1–1), (1–1), (1–1)	1	
8 (MHNCI 069)	1, 1, 1, 1, 1, 2, 2, 1, 2, 1, 1	2	
8 (MHNCI 070)	1, 1, 1, 1, 1, 1, 1, 2, 1, 2, 2, 2, 2, 2, 2, 2, 1	4	
9 (MHNCI 071)	1, 1, 1, 1, 1, 1, 2, 2, 2, 2, 2, 2, 2, 2, 2, 2, 2, 2, 2, 2, 2, (2–2), (2–2), 2, (2–2), 2, 2	1	
9 (MHNCI 072)	1, 1, 1, 1, 2, 2, 2, 2, 2, 2, 2, 2, 2, 2, 2, 2, (2–2), (2–2), 2, 2, (2–2)	0	
9 (MHNCI 073)	1, 1, 1, 1, 1, 1, 2, 2, 2, 2, 2, 2, 2, 2, 2, 2, 2, 2, 2, (2–2), 2, (2–2), (2–2), (2–2), 2, 2, 2	2	
9 (MHNCI 074)	1, 1, 1, 1, 1, 1, 2, 2, 2, 2, 2, 2, 2, 2, 2, 2, 2, 2, 2, (2–2), (2–2), 2, (2–2), 2, (2–2), (2–2)	2	
9 (MHNCI 075)	2, 2, 2, 2, 2, 2, 2, 2, 2, 2, 2, 2, 2	?	
10 (MHNCI 076)	2, 2, 1, (2–2), (2–2), (2–2)	?	
10 (MHNCI 077)	2, 2, 2, 2, 2, 2, 2, 2, 2, 2, 2, 2, (2–2), (2–2), (2–2), (2–2), (2–2), (2–2), (2–2), (2–2), (2–2)	1	
10 (MHNCI 078)	2, 2, 2, 2, 2, 2, 2, 2, (2–2), (2–2), (2–2), (2–2)	1	
11 (MHNCI 079)	1, 3, (2–2), (1–3), (3–3), (3–3), 1, (2–1), 3, (1–1), (2–1), (2–2), (2–2), (3–2), (2–3)	?	
12 (MHNCI 080)	1, 1, 1, 2, 2, 2, 2, 2, (2–2), (2–2), (2–2), (2–2)	1	
12 (MHNCI 081)	2, 2, 2, 2, (3–2), (2–2), (2–2), (3–3), (2–2), (2–2)	4	
12 (MHNCI 082)	1, 1, 1, 1, 1, 2, 2, 2, 2, (2–2), (2–2), (2–2)	1	
Note:

Pulses per note groups are indicated between parenthesis, but indicating separately by “–” the number of pulses in each particular note of the group. Remaining notes represent isolated notes.

Brachycephalus mirissimus emitted an advertisement call of 111.83 ± 46.60 s (37.70–255.20; Fig. 7A). An individual can remain silent for several minutes after emitting an advertisement call (occasionally for more than 20 min), when it emits a new call. Advertisement calls included 23.55 ± 10.29 notes (6–52), with a note rate of 11.69 ± 2.12 notes per minute (7.48–15.93). The advertisement calls included both isolated notes and note groups (in this case, with two notes involved in each particular note group; Table 3; Figs. 7C and 7E). Advertisement calls could be composed only by isolated notes (26.7% of the advertisement calls), but usually included both isolated notes and note groups (Table 3). Each advertisement call with note groups began with isolated notes and then changed to note groups (Table 2). Some advertisement calls began with warming notes, at least with up to three of this attenuated note (see example of one warming note in the oscillogram of the Fig. 7A—a small peak of energy just above 0). The part of the advertisement call composed of isolated notes has a duration of 63.05 ± 22.83 s (16.26–100.30) and include 13.85 ± 5.34 notes (6–25), emitted in a rate of 10.54 ± 1.59 note per minute (7.48–14.28). The part of the advertisement call composed by note groups have a duration of 47.06 ± 39.31 s (6.71–182.40) and include 14.00 ± 7.58 notes (6–38), emitted in a rate of 17.77 ± 4.66 note per minute (12.20–27.43). There are 1.69 ± 0.47 pulses per isolated notes (1–3; Figs. 7B and 7D) and 1.96 ± 0.32 pulse per note in note groups (1–3; Figs. 7C and 7E). In each note groups (two notes counting in a single value of pulses), there are 3.92 ± 0.60 pulses (2–6). A total of seven combinations of number of pulses in each note in note groups were recorded, that is, 1–1 (n = 10 note groups; n = 2 individuals), 2–2 (n = 135 note groups; n = 10 individuals), 3–3 (n = 3 note groups; n = 2 individuals), 1–3 (n = 1 note group), 2–1 (n = 2 note groups; n = 2 individuals), 2–3 (n = 1 note group), and 3–2 (n = 2 note groups; n = 2 individuals). Note duration of isolated notes is 0.02 ± 0.01 s (0.00–0.03) and note duration of note groups is 0.43 ± 0.04 s (0.36–0.59). The inter-note interval in isolated notes is 5.83 ± 1.35 s (3.92–10.62) and the inter-note group interval is 7.04 ± 1.18 s (5.32–10.93). The inter-note interval within note groups is 0.39 ± 0.03 s (0.35–0.49). The note dominant frequency is 6.66 ± 0.28 kHz (6.00–7.23). Finally, the highest frequency is 8.42 ± 043 kHz (7.26–10.06) while the lowest frequency is 4.26 ± 0.71 kHz (2.67–5.62).

Figure 7 Advertisement call of Brachycephalus mirissimus.

Example of an entire advertisement call and also notes of other advertisement calls of Brachycephalus albolineatus. (A) Entire advertisement call (MHNCI 059, voucher MHNCI 10795). (B, D) All examples observed of isolated notes, with one pulse in B (MHNCI 067) and two pulses in D (MHNCI 073, voucher MNHCI 10794). (C, E) Examples of note groups, with 1–1 pulses in C (MHNCI 067) and 2–2 pulses in E (MHNCI 082, voucher MHNCI 10797). Spectrograms are produced with a FFT size of 8192 points, Hann window, and overlap of 90% in A and FFT 512 points, Hann window, and overlap of 90% in B–E.

Etymology. The specific epithet mirissimus is a superlative of the Latin adjective mirus, which means wonderful, marvelous.

Habitat, abundance, and distribution. We recorded B. mirissimus calling throughout the day under the leaf litter, but with more intense vocal activity in the morning and later in the day. We did not hear the species throughout the study area, as it showed a patchy distribution, and it is not homogeneously abundant in these patches. In the patch where it appeared particularly abundant, we heard 14 males in 202 m2, resulting in one calling male per 14.5 m2.

The species is known from the type locality (Fig. 6), where it was found in a patchy distribution between 470 and 540 m a.s.l. in montane forest (Floresta Ombrófila Densa Montana; Fig. 8) that reaches about 18–28 m in height. We did not find the species in montane forest with a lower canopy (<10 m), which was in a very steep terrain. We estimate its “actual” extent of occurrence to be 56.8 ha (calculated excluding forested areas in very steep terrains). That estimate resulted in three in-line polygons, distant from each other by 190 and 60 m, which became isolated due to replacement of the original vegetation by Eucalyptus sp., Pinus sp. and palm plantations of Archontophoenix alexandrae H. Wendl. & Drude (Fig. 9; areas encompassed by polygons are 28.3, 23.1, and 5.4 ha). The historical extent of occurrence (i.e., before the deforestation) taken the above criteria, resulted in a continuous polygon of 111.8 ha.

Figure 8 Vegetation at the type locality of Brachycephalus mirissimus, at 535 m above sea level, characterized by high-elevation forest (Floresta Ombrófila Densa Montana).

(A) The middle and low strata of the forest, evidencing the habitat of Brachycephalus mirissimus in the leaf litter. (B) General view of the Morro Santo Anjo’s forest. Photographs by Luiz F. Ribeiro.

Figure 9 Impacts at the type locality of Brachycephalus mirissimus.

In (A) Morro Santo Anjo with plantation of Pinus sp. along the slope. In (B) forest boundary with palm plantation of Archontophoenix alexandrae. Photograph by Luiz F. Ribeiro.

We recorded some species at the type locality that are typical of high altitudes, such as Quelusia regia Vell., in the case of plants, and Piculus aurulentus, Attila phoenicurus, Carpornis cucullata, and Scytalopus speluncae (taxonomy according to Maurício et al. (2010)), in the case of birds. On the other hand, we also recorded some plants and birds at the type locality that are typical of lowland habitats, for example Bathysa australis (A.St.-Hil.) K.Schum., Cecropia sp., Bactris setosa Mart, and Euterpe edulis Mart., in the case of plants, and Myrmotherula unicolor, Conopophaga melanops, Hemitriccus orbitatus, and Phylloscartes kronei, in the case of birds, showing a mixed flora and fauna from both high and low elevations.

Remarks. The type locality of B. mirissimus is 17.4 km distant in a straight line from the type locality of B. albolineatus, 18.9 km distant from the type locality of B. fuscolineatus, and 19.5 km distant from the type locality of B. boticario.

Discussion

Our phylogenetic analysis provided clear support for B. mirissimus as part of the southernmost clade of Brachycephalus, which includes B. albolineatus, B. fuscolineatus, and B. boticario (Fig. 6). In particular, B. mirissimus was placed in a clade with B. albolineatus and B. fuscolineatus with moderate support, but there was not enough signal in the analysis to uncover the relationships within this clade. Despite the close phylogenetic and geographical proximity between these four species indicated (Fig. 6), they differ considerably in coloration. On the other hand, the calls of B. mirissimus are nearly indistinguishable from the only closely related Brachycephalus species whose call has been described to date—B. albolineatus (Bornschein et al., 2018). Such low rate of evolution in prezygotic isolation mechanisms is not unexpected given that these species are allopatric and the risk of hybridization is minimized.

Brachycephalus mirissimus is the fourth species of the B. pernix group whose advertisement call has been described to date (see Garey et al., 2012; Bornschein et al., 2018; Monteiro et al., 2018). The advertisement calls of species of B. pernix group share a similar overall resemblance, including structural, temporal, and spectral patterns (see Table S2). This similarity is most apparent when comparing calls of B. albolineatus and B. mirissimus by the presence of both isolated notes and note groups. On the other hand, note groups are absent from the calls of B. tridactylus, according to our analysis (see list of examined recordings in Appendix 2, which also includes the recording used in the original description by Garey et al. (2012)). The description of the call of B. actaeus under the call-centered approach (sensu Köhler et al., 2017; Table S2) prevents us from recognizing if this species presents or not note groups.

Although the number of pulses per note in the advertisement call of B. mirissimus and B. albolineatus is not a diagnostic character, given that both species present 1–3 pulses per note, some particularities are striking. In the new species, 20.6% of the notes showed one pulse (n = 138 notes), 77.5% showed two pulses (n = 521 notes), and only 1.9% showed three pulses (n = 13 notes), while in B. albolineatus there was a similar low percentage of notes with one pulse (4.7%; n = 26 notes), a relatively reduced percentage of notes with two pulses (46.4%; n = 257 notes), but a comparatively very high percentage of notes with three pulses (48.9%; n = 271 notes). The presence of three pulses in these species presents diametrically opposite results, which may reflect a tendency of the new species to be losing the emission of notes with three pulses.

The altitudinal range of occurrence of B. mirissimus confirms the tendency of species of B. pernix group to occur at lower altitudes when at higher latitudes (Bornschein et al., 2016b). The abundance of Brachycephalus, estimated using the same methodology, revealed that B. mirissimus is much less abundant than B. curupira and B. albolineatus. We obtained the estimates of one calling individual of B. mirissimus per 14.5 m2 (this study), one calling individual of B. curupira per two to three m2 (Ribeiro et al., 2017), and one calling individual of B. albolineatus per 3–4 m2 (Bornschein et al., 2016b). These results raise concern for the conservation of B. mirissimus because all of those estimates were made where the species appeared to be most abundant, yet its abundance is about four to six times lower than its congeners (see below).

The present extent of occurrence of B. mirissimus (56.8 ha) is among the smallest ranges of vascular plants and fishes around the world (<100 ha; Brown, Stevens & Kaufman, 1996) and is similar to other microendemic Brachycephalus from southern Brazil (Bornschein et al., 2016a). According to the criteria for classification of endangered species of the world (IUCN, 2012), B. mirissimus corresponds to Critically Endangered based on the following criteria: B2a, b(i). Evidence for adherence to this criterion includes selective harvest of trees, edge effects, and erosion around two roads that go across one of the forest fragments that constitute the extent of occurrence of the new species. We estimate that the present extent of occurrence was reduced by 53% by deforestation (Fig. 9).

Appendix 1. Examined Specimens

Brachycephalus actaeus. SANTA CATARINA: Morro do Cantagalo, municipality of São Francisco do Sul MHNCI 10829–31, and seven additional uncatalogued specimens.

Brachycephalus albolineatus. SANTA CATARINA: Morro Boa Vista, boundary of the municipalities of Jaraguá do Sul and Massaranduba MHNCI 10290 (holotype), MHNCI 10295–10300, MNRJ 90349 (all paratypes).

Brachycephalus alipioi. ESPÍRITO SANTO: Alto Castelinho, municipality of Vargem Alta MHNCI 10804–6.

Brachycephalus auroguttatus. SANTA CATARINA: Pedra da Tartaruga, municipality of Garuva DZUP 375 (holotype), DZUP 373–4, 376–85, 387–89 (all paratypes).

Brachycephalus boticario. SANTA CATARINA: Morro do Cachorro, boundary of the municipalities of Blumenau, Gaspar, and Luiz Alves DZUP 440 (holotype), DZUP 414–5, 438–9, 444–5, 459 (all paratypes).

Brachycephalus brunneus. PARANÁ: Camapuã, Serra dos Órgãos, boundary of the municipalities of Campina Grande do Sul and Antonina MHNCI 10165–74; Caratuva, Serra dos Órgãos, municipality of Campina Grande do Sul MHNCI 1919–20, MNRJ 40289–91 (paratypes), MHNCI 10175–84.

Brachycephalus coloratus. PARANÁ: Estância Hidroclimática Recreio da Serra, Serra da Baitaca, municipality of Piraquara MHNCI 10273 (holotype), MHNCI 10274–79, MNRJ 89949–50 (all paratypes).

Brachycephalus curupira. PARANÁ: Serra do Salto, Malhada District, municipality of São José dos Pinhais MHNCI 10280 (holotype), MHNCI 10281–87, 10292 (all paratypes).

Brachycephalus didactylus. RIO DE JANEIRO: municipality of Engenheiro Paulo de Frontin ZUEC 10825; Sacra Família do Tinguá, municipality of Engenheiro Paulo de Frontin ZUEC 1132–3, MZUSP 13613–20, 64810–1, 94621 (topotypes).

Brachycephalus ephippium. RIO DE JANEIRO: Parque Nacional Serra dos Órgãos MZUSP 104140–7; Vale de Revolta MCZ A–108655. SÃO PAULO: municipality of Cotia MHNCI 2611–16.

Brachycephalus ferruginus. PARANÁ: Olimpo, Serra do Marumbi, municipality of Morretes MHNCI 125, 128 (topotypes), MHNCI five specimens uncatalogued.

Brachycephalus fuscolineatus. SANTA CATARINA: Morro do Baú, municipality of Ilhota DZUP 159 (holotype), DZUP 158, 160, 401–5 (all paratypes), MHNCI two specimens uncatalogued.

Brachycephalus hermogenesi. SÃO PAULO: Picinguaba, Parque Estadual da Serra do Mar, municipality of Ubatuba ZUEC 9715 (holotype), ZUEC 9716–25 (paratypes), MHNCI 10823–25.

Brachycephalus izecksohni. PARANÁ: Torre da Prata, Serra da Prata, boundary of the municipalities of Morretes, Paranaguá, and Guaratuba CFBH 7381–2, 7384 (all paratypes), MHNCI 10835.

Brachycephalus leopardus. PARANÁ: Morro dos Perdidos, municipality of Guaratuba DZUP 274–83, MHNCI three specimens uncatalogued; Serra do Araçatuba, municipality of Tijucas do Sul DZUP 490 (holotype), DZUP 478–89, 491–2 (all paratypes).

Brachycephalus mariaeterezae. SANTA CATARINA: Reserva Particular do Patrimônio Natural Caetezal, top of the Serra Queimada, municipality of Joinville MHNCI 9811 (holotype), DZUP 372, 393–9 (all paratypes).

Brachycephalus nodoterga. SÃO PAULO: Reserva Biológica Tamboré, municipality of Santana de Parnaíba MZUSP 147711–6.

Brachycephalus olivaceus. SANTA CATARINA: base of the Serra Queimada, municipality of Joinville MHNCI 9813 (holotype), DZUP 371 (paratype); Castelo dos Bugres, municipality of Joinville MHNCI 9814–8 (paratypes); Morro do Boi, municipality of Corupá MHNCI 10288–9.

Brachycephalus pernix. PARANÁ: Anhangava, Serra da Baitaca, municipality of Quatro Barras MNRJ 17349 (holotype), CFBH 2597–8 (paratypes), MHNCI 1818–9, 3000–4 (all paratypes), MHNCI 1820, ZUEC 9433–7 (paratypes), DZUP 539–55.

Brachycephalus pitanga. SÃO PAULO: SP 125, municipality of São Luiz do Paraitinga MHNCI 10733–34; Trilha do Ipiranga 50 m from the Rio Ipiranga, Núcleo Santa Virgínia, Parque Estadual da Serra do Mar, municipality of São Luiz do Paraitinga MHNCI 10821–22.

Brachycephalus pombali. PARANÁ: Morro dos Padres, Serra da Igreja, municipality of Morretes CFBH 8042 (holotype), 8043–53 (paratypes), MHNCI 11 specimens uncatalogued.

Brachycephalus quiririensis. SANTA CATARINA: Serra do Quiriri, municipality of Campo Alegre DZUP 172 (holotype), DZUP 171, 173–6, 524–30 (all paratypes), MHNCI five specimens uncatalogued.

Brachycephalus sulfuratus. SÃO PAULO: base of the Serra Água Limpa, municipality of Apiaí DZUP 362; near Jurupará dam, municipality of Piedade MHNCI 10829–31. PARANÁ: Caratuval, near the Parque Estadual das Lauráceas, municipality of Adrianópolis DZUP 139; Corvo, municipality of Quatro Barras DZUP 150–7; Fazenda Thalia, municipality of Balsa Nova DZUP 221–4; Mananciais da Serra, municipality of Piraquara MHNCI 10302; Recanto das Hortências, municipality of São José dos Pinhais DZUP 463; Salto do Inferno, Rio Capivari, municipality of Bocaiúva do Sul MHNCI 9800. SANTA CATARINA: Morro do Garrafão, municipality of Corupá MHNCI 10826–28.

Brachycephalus toby. SÃO PAULO: Morro do Corcovado, Parque Estadual da Serra do Mar, municipality of Ubatuba MHNCI 10807–09.

Brachycephalus verrucosus. SANTA CATARINA: Morro da Tromba, municipality of Joinville MHNCI 9819 (holotype), MHNCI 9820, DZUP 464–78 (all paratypes).

Brachycephalus tridactylus. PARANÁ: Serra do Morato, Reserva Natural Salto Morato, municipality of Guaraqueçaba MHNCI 10185–89, MHNCI 10294, 10729–30 (topotypes).

Brachycephalus vertebralis. RIO DE JANEIRO/SÃO PAULO: Morro Cuscuzeiro, Núcleo Picinguaba of the Parque Estadual da Serra do Mar and Parque Nacional da Serra da Bocaina, boundary of the municipalities of Parati, Rio de Janeiro state, and Ubatuba, São Paulo state MHNCI 10810–20.

Appendix 2. Examined Recordings

Brachycephalus albolineatus. SANTA CATARINA: Morro Boa Vista, boundary of the municipalities of Jaraguá do Sul and Massaranduba MHNCI 001–34.

Brachycephalus tridactylus. PARANÁ: Serra do Morato, Reserva Natural Salto Morato, municipality of Guaraqueçaba MHNCI 035–51, FNJV 0032950.

Supplemental Information

Supplemental Information 1 Raw data: Molecular dataset for Brachycephalus mirissimus.

Click here for additional data file.

Supplemental Information 2 Table S1. Accession numbers of the sequences used in the present study.

Click here for additional data file.

Supplemental Information 3 Table S2. Comparison of the features used to describe the advertisement call of Brachycephalus.

Click here for additional data file.

Supplemental Information 4 Advertisement call of Brachycephalus mirissimus.

052; Brachycephalus mirissimus; 18 jan 2018; individual 01, example 01; A. E. Confetti; not collected.

Click here for additional data file.

Supplemental Information 5 Advertisement call of Brachycephalus mirissimus.

053; Brachycephalus mirissimus; 18_jan_2018; individual 01, example 02; A. E. Confetti; not collected.

Click here for additional data file.

Supplemental Information 6 Advertisement call of Brachycephalus mirissimus.

054; Brachycephalus mirissimus; 18_jan_2018; individual 02, example 01; A. E. Confetti; not collected.

Click here for additional data file.

Supplemental Information 7 Advertisement call of Brachycephalus mirissimus.

055; Brachycephalus mirissimus; 18_jan_2018; individual 02, example 02; A. E. Confetti; not collected.

Click here for additional data file.

Supplemental Information 8 Advertisement call of Brachycephalus mirissimus.

056; Brachycephalus mirissimus; 18_jan_2018; individual 03; two initial notes missing; A. E. Confetti; not collected.

Click here for additional data file.

Supplemental Information 9 Advertisement call of Brachycephalus mirissimus.

057; Brachycephalus mirissimus; 18_jan_2018; individual 04, example 01; A. E. Confetti; collected MHNCI 10795.

Click here for additional data file.

Supplemental Information 10 Advertisement call of Brachycephalus mirissimus.

058; Brachycephalus mirissimus; 18_jan_2018; individual 04, example 02; A. E. Confetti; collected MHNCI 10795.

Click here for additional data file.

Supplemental Information 11 Advertisement call of Brachycephalus mirissimus.

059; Brachycephalus mirissimus; 18_jan_2018; individual 04, example 03; A. E. Confetti; collected MHNCI 10795.

Click here for additional data file.

Supplemental Information 12 Advertisement call of Brachycephalus mirissimus.

060; Brachycephalus mirissimus; 18_jan_2018; individual 04, example 04; A. E. Confetti; collected MHNCI 10795.

Click here for additional data file.

Supplemental Information 13 Advertisement call of Brachycephalus mirissimus.

061; Brachycephalus mirissimus; 18_jan_2018; individual 05, example 01; fist note missing; L. F. Ribeiro; not collected.

Click here for additional data file.

Supplemental Information 14 Advertisement call of Brachycephalus mirissimus.

062; Brachycephalus mirissimus; 18_jan_2018; individual 05, example 02; first note missing; L. F. Ribeiro; not collected.

Click here for additional data file.

Supplemental Information 15 Advertisement call of Brachycephalus mirissimus.

063; Brachycephalus mirissimus; 18_jan_2018_individual 05, example 03; fist note missing; L. F. Ribeiro; not collected.

Click here for additional data file.

Supplemental Information 16 Advertisement call of Brachycephalus mirissimus.

064; Brachycephalus mirissimus; 18_jan_2018; individual 05, example 04; fist note missing; L. F. Ribeiro; not collected.

Click here for additional data file.

Supplemental Information 17 Advertisement call of Brachycephalus mirissimus.

065; Brachycephalus mirissimus; 18_jan_2018; individual 05, example 05; two initial notes missing; L. F. Ribeiro; not collected.

Click here for additional data file.

Supplemental Information 18 Advertisement call of Brachycephalus mirissimus.

066; Brachycephalus mirissimus; 18_jan_2018; individual 06, ex_01; three or four initial notes missing; L. F. Ribeiro; not collected.

Click here for additional data file.

Supplemental Information 19 Advertisement call of Brachycephalus mirissimus.

067; Brachycephalus mirissimus; 18_jan_2018; individual 07; first note missing; L. F. Ribeiro; not collected.

Click here for additional data file.

Supplemental Information 20 Advertisement call of Brachycephalus mirissimus.

068; Brachycephalus mirissimus; 15_jan_2018; individual 08, example 01 plus individual 09, example 01; recorded from 3-5 m of the emitter; initial half missing from both individual; collected MHNCI 10793 (individual 8), MHNCI 10974 (individual 9); M. R. Bornschein.

Click here for additional data file.

Supplemental Information 21 Advertisement call of Brachycephalus mirissimus.

069; Brachycephalus mirissimus; 15_jan_2018; individual 08, example 02, recorded from 150 cm of the emitter; two initial notes missing; with wind; M. R. Bornschein; collected MHNCI 10793.

Click here for additional data file.

Supplemental Information 22 Advertisement call of Brachycephalus mirissimus.

070; Brachycephalus mirissimus; 15_jan_2018; individual 08, example 03; recorder from 30-100 cm of the emitter; four initial notes missing; with wind; M. R. Bornschein; collected MHNCI 10793.

Click here for additional data file.

Supplemental Information 23 Advertisement call of Brachycephalus mirissimus.

071; Brachycephalus mirissimus; 15_jan_2018; individual 09, example 02; recorded from 150 cm of the emitter; first note missing; with wind; M.R. Bornschein; collected MHNCI 10794.

Click here for additional data file.

Supplemental Information 24 Advertisement call of Brachycephalus mirissimus.

072; Brachycephalus mirissimus; 15_jan_2018; individual 09, example 03; recorded from 30 cm of the emitter; with wind; M. R. Bornschein; collected MHNCI 10794.

Click here for additional data file.

Supplemental Information 25 Advertisement call of Brachycephalus mirissimus.

073; Brachycephalus mirissimus; 15_jan_2018; individual 09, example 04; recorded from 20-50 cm of the emitter; two initial notes missing; with wind; M. R. Bornschein; collected MHNCI 10794.

Click here for additional data file.

Supplemental Information 26 Advertisement call of Brachycephalus mirissimus.

074; Brachycephalus mirissimus; 15_jan_2018; individual 09, example 05; recorded from 20-50 cm of the emitter; two initial notes missing; with wing; M. R. Bornschein; collected MHNCI 10794.

Click here for additional data file.

Supplemental Information 27 Advertisement call of Brachycephalus mirissimus.

075; Brachycephalus mirissimus; 15_jan_2018; individual 09; example 06; recorded from 10-30 cm of the emitter; initial half missing; with wind; M. R. Bornschein; collected MHNCI 10794.

Click here for additional data file.

Supplemental Information 28 Advertisement call of Brachycephalus mirissimus.

076; Brachycephalus mirissimus; 18_jan_18; individual 10, example 01; recorded from 2 m of the emitter; M. R. Bornschein; collected MHNCI 10798.

Click here for additional data file.

Supplemental Information 29 Advertisement call of Brachycephalus mirissimus.

077; Brachycephalus mirissimus; 18_jan_18; individual 10, example 02; recorded from 50 cm of the emitter; first note missing; M. R. Bornschein; collected MHNCI 1079.

Click here for additional data file.

Supplemental Information 30 Advertisement call of Brachycephalus mirissimus.

078; Brachycephalus mirissimus; 18_jan_18; individual 10, example 03; recorded from 50-150 cm of the emitter; first note missing; M. R. Bornschein; collected MHNCI 10798.

Click here for additional data file.

Supplemental Information 31 Advertisement call of Brachycephalus mirissimus.

079; Brachycephalus mirissimus; 18_jan_18; individual 11; recorded from 2 m of the emitter; M. R. Bornschein; collected MHNCI 10796.

Click here for additional data file.

Supplemental Information 32 Advertisement call of Brachycephalus mirissimus.

080; Brachycephalus mirissimus; 18_jan_18; individual 12, example 01; recorded from 20-50 cm of the emitter; first note missing; M. R. Bornschein; collected MHNCI 10797.

Click here for additional data file.

Supplemental Information 33 Advertisement call of Brachycephalus mirissimus.

081; Brachycephalus mirissimus; 18_jan_18; individual 12, example 02; recorded from 50 cm of the emitter; four initial notes missing; M. R. Bornschein; collected MHNCI 10797.

Click here for additional data file.

Supplemental Information 34 Advertisement call of Brachycephalus mirissimus.

082; Brachycephalus mirissimus; 18_jan_18; individual 12, example 03; recorded from 20-40 cm of the emitter; first note missing; M. R. Bornschein; collected MHNCI 10797.

Click here for additional data file.

We thank the followings curators for facilitating our access to examined specimens, namely Julio Cesar Moura Leite (MHNCI), José P. Pombal Jr. (MNRJ), and José Rosado (MCZ). Luís Felipe de Toledo provided access to recordings at FNJV. Larissa Teixeira and Adriane Smythe helped in field work. Liliane Pires helped in formatted tables and figures. Drawings were made by Verônica R. Apolônio. We thank Tânia M. Costa for the access to a stereoscopic microscope used for drawing the holotype.

Additional Information and Declarations

Competing Interests

Author Contributions

Field Study Permissions

DNA Deposition

Data Availability

New Species Registration

Marcio R. Pie is an Academic Editor for PeerJ.

Marcio R. Pie conceived and designed the experiments, performed the experiments, analyzed the data, contributed reagents/materials/analysis tools, prepared figures and/or tables, authored or reviewed drafts of the paper.

Luiz F. Ribeiro conceived and designed the experiments, performed the experiments, analyzed the data, contributed reagents/materials/analysis tools, prepared figures and/or tables, authored or reviewed drafts of the paper.

André E. Confetti performed the experiments, analyzed the data, contributed reagents/materials/analysis tools, prepared figures and/or tables.

Mário J. Nadaline performed the experiments, analyzed the data, contributed reagents/materials/analysis tools.

Marcos R. Bornschein conceived and designed the experiments, performed the experiments, analyzed the data, contributed reagents/materials/analysis tools, prepared figures and/or tables, authored or reviewed drafts of the paper, approved the final draft.

The following information was supplied relating to field study approvals (i.e., approving body and any reference numbers):

Collection permits for this study were issued by ICMBIO (10.500 and 55918–1).

The following information was supplied regarding the deposition of DNA sequences:

16S rRNA sequences described here are available at GenBank MH136570 and in the Supplemental File.

The following information was supplied regarding data availability:

The audio files and gene sequences are available as Supplemental Files.

The following information was supplied regarding the registration of a newly described species:

Publication LSID: urn:lsid:zoobank.org:pub:AAB0DCA9-4587-42B3-812E-7752EB58F726

Species name registration (Brachycephalus mirissimus): urn:lsid:zoobank.org:act:B1C3F142-E68C-4B4F-9647-F69CA6C25EB9.

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
