# Peer review of "A new species of Brachycephalus (Anura: Brachycephalidae) from southern Brazil"

_PeerJ, doi:10.7717/peerj.5683_

## Round 0.1 · original submission · Major Revisions

The reviewers have highlighted the strengths of you submitted manuscript and proposed changes to improve the quality and clarity of the paper. The English needs additional efforts in editing. Please refer to the changes suggested by reviewers 2 and 3, including the comments/corrections in the annotated manuscript. I feel that it is also necessary to change the title. In the current version the use of the term 'remarkable' is not justified, based on the characteristics provided in the manuscript.

Reviewer 1 ·

Basic reporting

The report is absolutely clear and certainly adheres to the standards expected in modern taxonomy and systematic biology.

Experimental design

The research problem is well defined in the context of the discovery and description of biodiversity, and the methods employed by the Authors are those needed to uncover the unknown biodiversity.

Validity of the findings

The Authors conclusively demonstrate that the organism of interest in new to science.

Additional comments

Pie and Coauthors describe a new species of Brachycephalus from southern Brazil. The Authors have produced an outstanding, detailed description of this new taxon from phenotypical, ecological, and genealogical standpoints. I have nothing to add except for asking one question. In what respect is the new species remarkable? Remarkable means "worthy of being or likely to be noticed especially as being uncommon or extraordinary". What mechanisms, processes or the outcomes of mechanisms and processes render this new species extraordinary? I fear that by using the word "remarkable" the Authors may have placed upon themselves the burden to justify the use of this word on scientific grounds.

·

Basic reporting

There are numerous minor problems with the usage of the English language throughout the manuscript, especially with the usage of prepositions. The manuscript structure is good, but I suggest to delete the "conclusions" paragraph which seems superfluous.

Experimental design

no comment

Validity of the findings

no comment

Additional comments

The authors make a strong point for their suggested taxonomy including the description of a new species which is well supported by coloration, morphology, DNA barcoding data, and bioacoustics. I have made numerous minor edits and several comments/suggestions in the attached PDF. I am looking forward to seeing this work in print.

·

Basic reporting

The paper wish to describe a new species of Brachycephalus belonging to the B. pernix group and the text is mostly clear and unambiguous, yet some improvements could be made to sintax and grammar.

The authors have a tendency to cite themselves along the text, sometimes ignoring more consecrated references from various disciplines. In some cases, this is not a problem, although other cases are more problematic and were highlighted along the text.

Article structure is generally ok. However some topics need major changes (detailed in the text). Some figures have low quality and also demand adjustments, specially the photos of holotype.

The article is self-contained, yet the results presented are not enough to corroborate the hypothesis of a new species.

Experimental design

The paper is within the Aims and Scope of the journal.

The research question is well defined, relevant and meaningful.

Although the research included highly technical methods, sometimes materials and references are not enough to fulfill these analysis, which led to dubious or incomparable results. Some critical points are highlighted along the text.

Validity of the findings

Although the candidate species present some features that could distinguish it from its congeners, the overall result is inconclusive, with not enough evidence to corroborate the hypothesis of a new species.

Some of the data is robust (e.g. external morphology data), but other data require major corrections that could impact the results (e.g. bioacoustics).

Ultimately, the work's conclusion is within the scope and dialogs with the remaining text.

---

## Round 0.2 · accepted · Accept

Thank you very much for the thorough revision of the manuscript and the consideration of the reviewers' comments.

# ·

Basic reporting

no further comments

Experimental design

no further comments

Validity of the findings

no further comments

Additional comments

The authors produced a fine revision of their work addressing the corrections and comments of the reviewers. I liked the original manuscript and am happy to see this improved revision. I am looking forward to see this work being published. Gunther Köhler